# Intersectional, anterograde transsynaptic targeting of neurons receiving monosynaptic inputs from two upstream regions

Takuma Kitanishi [1,2✉], Mariko Tashiro[1], Naomi Kitanishi[1] & Kenji Mizuseki [1✉]

A brain region typically receives inputs from multiple upstream areas. However, currently, no method is available to selectively dissect neurons that receive monosynaptic inputs from two upstream regions. Here, we developed a method to genetically label such neurons with a single gene of interest in mice by combining the anterograde transsynaptic spread of adeno-associated virus serotype 1 (AAV1) with intersectional gene expression. Injections of AAV1 expressing either Cre or Flpo recombinases and the Cre/Flpo double-dependent AAV into two upstream regions and the downstream region, respectively, were used to label post-synaptic neurons receiving inputs from the two upstream regions. We demonstrated this labelling in two distinct circuits: the retina/primary visual cortex to the superior colliculus and the bilateral motor cortex to the dorsal striatum. Systemic delivery of the intersectional AAV allowed the unbiased detection of the labelled neurons throughout the brain. This strategy may help analyse the interregional integration of information in the brain.

[1] Department of Physiology, Osaka City University Graduate School of Medicine, Osaka 545-8585, Japan. [2] PRESTO, Japan Science and Technology Agency (JST), Kawaguchi, Saitama 332-0012, Japan. ✉email: kitanishi.takuma@med.osaka-cu.ac.jp; mizuseki.kenji@med.osaka-cu.ac.jp

The brain consists of many distinct regions with corresponding functions. A single brain region generally receives synaptic inputs from multiple upstream areas and distributes the information processed within the region to multiple downstream regions. Such integration and distributions are fundamental interregional interactions that support a variety of brain functions[1,2]. The neurons that distribute information to two or more downstream areas through collateral projections can be readily labelled using multiple retrograde tracers[3,4]. Moreover, one can genetically target these neurons to monitor and manipulate their activity using several approaches, such as the dual retrograde viral vector infection[5] and optogenetic identification of axonal projections during extracellular recordings[6–9]. In contrast, although a range of viral vectors and transgenic animals has been successfully used to genetically target specific circuit structures[10–12], no method is currently available to selectively express a single gene of interest in neurons that receive monosynaptic inputs from multiple upstream regions, thereby limiting the study of the structure and function of such neurons.

Here, we aimed to develop an intersectional, anterograde transsynaptic targeting approach that allows for specific genetic labelling of neurons that receive monosynaptic inputs from two upstream regions. This approach combines adeno-associated virus serotype 1 (AAV1)[13,14] with the previously developed intersectional expression system (INTRSECT)[15,16]. High titres of AAV1 exhibit anterograde transneuronal spread, which depends on neurotransmitter release machinery[13,14]. Widely applied in mice[17–22] and rats[23–25], injection of the Cre recombinase-expressing AAV1 and AAV with a Cre-inducible expression cassette into brain regions containing presynaptic and postsynaptic neurons, respectively, allows the selective labelling of postsynaptic neurons receiving synaptic input from the presynaptic region. INTRSECT is an intersectional gene expression system that depends on multiple recombinases[15,16]. The $C_{on}/F_{on}$ expression cassette of the INTRSECT system turns on gene expression only when both Cre and Flpo recombinases are co-expressed in the target cells. We hypothesised that introducing the AAV1 expressing Cre and Flpo into two upstream brain regions and a $C_{on}/F_{on}$ expression cassette-containing AAV into the downstream region may aid in selectively labelling neurons that receive monosynaptic inputs from both upstream regions (Fig. 1a). As a proof-of-principle, we aimed to demonstrate that postsynaptic neurons receiving both monosynaptic inputs can be selectively labelled in two distinct circuits: the retina/primary visual cortex (V1) to the superior colliculus (SC) and the bilateral secondary motor cortex (M2) to the dorsal striatum (DS) pathways. The labelling exhibited synaptic specificity with successful application in the two circuits. We also demonstrated that the

systemic delivery of the $C_{on}/F_{on}$ cassette is useful for the unbiased detection of neurons throughout the brain.

## Results

### Anterograde transneuronal labelling with AAV1-Cre and AAV1-Flpo.
To perform intersectional, anterograde transsynaptic targeting in the brain (Fig. 1a), we first tested site-specific recombination using Cre and Flpo recombinases in vitro. As the Cre- and Flpo-mediated recombination of the $C_{on}/F_{on}$ cassette has been well-characterised[15,16], this test was primarily conducted to verify the selective recombination with the newly constructed pAAV-hSyn-Flpo-3×FLAG plasmid, which expresses 3×FLAG-tagged Flpo. To do this, 293T cells were transfected with recombinase-expressing plasmids (pAAV-hSyn-Cre or pAAV-hSyn-Flpo-3×FLAG) and recombinase-inducible, enhanced yellow fluorescent protein (EYFP)-expressing plasmids (pAAV-EF1α-DIO-EYFP or pAAV-EF1α-fDIO-EYFP). Cre/DIO and Flpo/fDIO combinations selectively induced EYFP expression (Fig. 1b), confirming no cross-reactivity between Cre and Flpo-3×FLAG. For the $C_{on}/F_{on}$ plasmid (pAAV-hSyn-$C_{on}/F_{on}$-EYFP), unlike the cells transfected with the Cre- or Flpo-expressing plasmid alone (Fig. 1c), EYFP was only expressed in cells in which Cre- and Flpo-expressing plasmids were co-transfected (Fig. 1c), verifying the intersectional gene expression associated with the $C_{on}/F_{on}$ cassette[15].

We then tested the anterograde transneuronal spread of AAV1 expressing either Cre or Flpo under the hSyn promoter (hereafter, AAV1-Cre and AAV1-Flpo, respectively). AAV1 exhibits anterograde transneuronal spread in many, but not all, pathways[13,14,17–21,23–26]. Therefore, in the absence of published evidence, verifying the transneuronal spread in the pathway of interest is crucial. In addition to the expected anterograde transneuronal spread of AAV1, it can be transported in a retrograde direction from axons[13,27,28], potentially hampering the exclusive detection of cells labelled by anterograde transneuronal transport in the presence of a reciprocal connection between the targeted pre- and post-synaptic regions. Hence, we selected unidirectional pathways (V1 to ipsilateral SC, retina to contralateral SC, and M2 to contralateral DS) without back projection[29–34]. We injected either AAV1-Cre and AAV1-EF1α-DIO-EYFP or AAV1-Flpo and AAV1-EF1α-fDIO-EYFP locally into the presynaptic and postsynaptic regions, respectively (Fig. 2a). In all three tested pathways, EYFP-positive somata were densely observed in the postsynaptic regions (i.e., SC or DS) for both AAV1-Cre and AAV1-Flpo (Fig. 2b–g). EYFP expression in the SC and DS was absent when either the AAV1-Cre or AAV1-Flpo injection into the presynaptic region was omitted ($n = 4$ mice), confirming

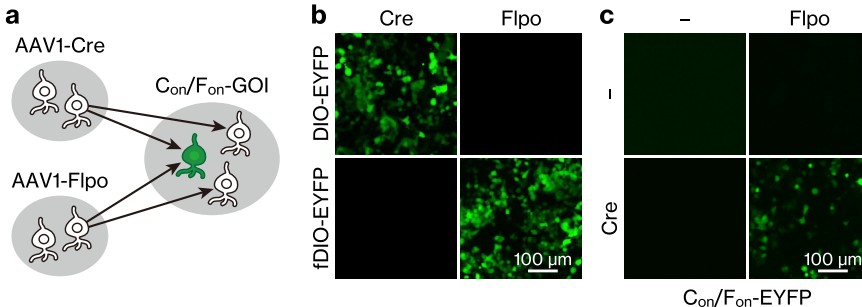

**Fig. 1 Outline of the intersectional, anterograde transsynaptic targeting system. a** Schematic of the intersectional, anterograde transsynaptic targeting of neurons that receive monosynaptic inputs from two upstream brain regions. **b** EYFP fluorescence in 293T cells transfected with recombinase (pAAV-hSyn-Cre or pAAV-hSyn-Flpo-3×FLAG) and recombinase-dependent (pAAV-EF1α-DIO-EYFP or pAAV-EF1α-fDIO-EYFP) plasmid combinations. **c** EYFP fluorescence in 293T cells transfected with Cre, Flpo, and $C_{on}/F_{on}$ plasmid combinations. The pAAV-hSyn-$C_{on}/F_{on}$-EYFP plasmid was transfected into all wells. AAV1 adeno-associated virus serotype 1, GOI gene of interest, EYFP enhanced yellow fluorescent protein.

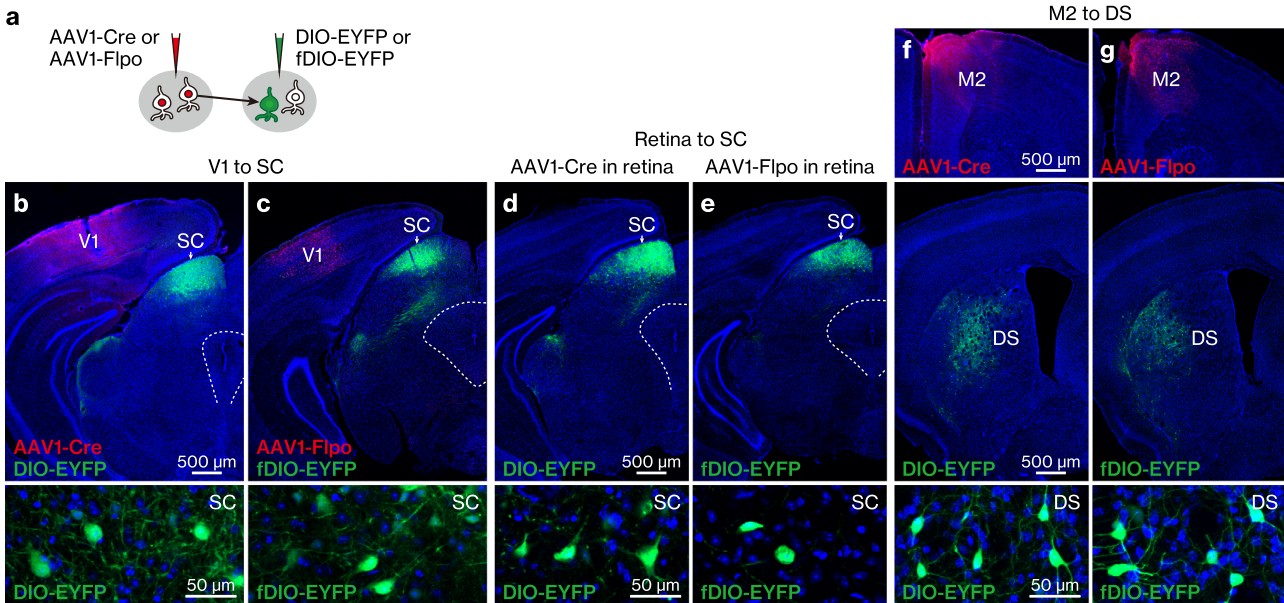

**Fig. 2 Anterograde transneuronal spread of AAV1-Cre and AAV1-Flpo. a** Schematic of the anterograde transneuronal labelling. **b–g** Anterograde transneuronal spread of AAV1-Cre (**b**, **d**, **f**) and AAV1-Flpo (**c**, **e**, **g**) from the V1 to ipsilateral SC (**b**, **c**), retina to contralateral SC (**d**, **e**), and M2 to contralateral DS (**f**, **g**) pathways. The Cre (red) and 3×FLAG-tagged Flpo (red) were detected with anti-Cre and anti-FLAG antibodies, respectively. All images represent coronal sections, and the bottom images show the labelled EYFP (green)-positive somata in the SC (**b–e**) and DS (**f**, **g**). Blue, DAPI. $N = 2$ (**b–e**, **g**) and 3 (**f**) mice under each experimental condition. The numbers of EYFP-positive cells per section in either the SC (**b–e**) or DS (**f**, **g**) were as follows: 32.2 ± 41.8 (**b**), 43.5 ± 0.7 (**c**), 35.6 ± 7.0 (**d**), 27.3 ± 38.5 (**e**), 87.4 ± 10.0 (**f**), and 71.3 ± 10.0 (**g**). AAV1 adeno-associated virus serotype 1, EYFP enhanced yellow fluorescent protein, V1 primary visual cortex, SC superior colliculus, M2 secondary motor cortex, DS dorsal striatum.

recombinase-dependent EYFP expression. These results suggest that both AAV1-Cre and AAV1-Flpo exhibited anterograde transneuronal spread, inducing site-specific recombination in postsynaptic neurons in all pathways tested in this study. To the best of our knowledge, this is the first demonstration of the transneuronal spread of AAV1-Cre in the M2-DS pathway and AAV1-Flpo in the retina-SC and M2-DS pathways.

**Intersectional, anterograde transsynaptic labelling in the retina/V1 to the superficial SC (sSC) pathway.** To genetically target neurons that receive monosynaptic inputs from two upstream regions, we performed intersectional, anterograde transneuronal labelling. The sSC receives synaptic inputs from both the retina and V1[35,36], but whether individual sSC neurons receive monosynaptic inputs from both regions remains elusive[32,37]. We injected AAV1-Cre, AAV1-Flpo, and AAV1-hSyn-$C_{on}/F_{on}$-EYFP into the right eye, left V1, and left SC, respectively (Fig. 3a). Together with Cre and Flpo expression in the injected regions (Fig. 3b, c), many EYFP-positive neurons were observed in the left sSC but not in the adjacent regions (Fig. 3c, d). EYFP expression was absent when either AAV1-Cre or AAV1-Flpo injection was omitted ($n = 4$ mice each), indicating that the injection of both vectors into the presynaptic regions (i.e., retina and V1) was necessary for EYFP labelling in the postsynaptic area (i.e., sSC).

Next, we aimed to achieve unbiased detection of the input-integrating neurons throughout the brain via the brain-wide delivery of the $C_{on}/F_{on}$-EYFP expression cassette. To this end, we used AAV-PHP.eB, which crosses the blood-brain barrier and transduces the whole brain following intravenous administration[38]. We injected AAV1-Cre and AAV1-Flpo locally into the right eye and left V1, respectively, as well as AAV-PHP.eB-hSyn-$C_{on}/F_{on}$-EYFP intravenously via the retroorbital sinus. EYFP-positive cells were observed in the left sSC (Fig. 3e) and left ventral lateral geniculate nucleus (LGNv) (Fig. 3f).

The LGNv does not demonstrate any back projection to the retina or V1[29–32], indicating that the observed EYFP-positive cells in the LGNv were likely the neurons integrating inputs from the retina and V1.

Although neurons in the retina and V1 project to the sSC, those in the auditory cortex (AC) mainly project to the deep SC (dSC)[35]. Taking advantage of these anatomically close but segregated pathways, we examined the synaptic specificity of the intersectional, anterograde transneuronal labelling. Consistent with the known AC-to-dSC projection, the injection of the AAV1-Flpo and AAV1-hSyn-fDIO-EYFP into the left AC and left SC, respectively, resulted in the preferential expression of EYFP in the left dSC (Fig. 3g, h). Next, we injected AAV1-Cre, AAV1-Flpo, and AAV1-hSyn-$C_{on}/F_{on}$-EYFP into the right eye, left AC, and left SC, respectively (Fig. 3i). As the retina and AC neurons innervate mostly distinct SC layers with minor overlap[30,39], if AAV1 spreads selectively to mono-synaptically connected SC neurons, this injection would label almost no neurons in the SC. Following AAV injection, EYFP-positive somata were rarely observed in the SC (Fig. 3j), and their number was significantly smaller than that labelled in the retina/V1 to SC pathway experiment (Fig. 3k and Supplementary Data 1). This result suggests that anterograde transneuronal transport was selective for labelling expected postsynaptic SC neurons.

**Intersectional, anterograde transsynaptic labelling in the bilateral M2 to DS pathway.** Next, we examined whether intersectional, anterograde transsynaptic labelling can be applied to another circuit, the bilateral M2 to DS pathway. The DS receives synaptic inputs from both the left and right M2 with no back projection[31,33,34]. The striatum is composed of medium spiny neurons, which are DARPP-32-positive principal cells comprising ~95% of the total striatal neuronal population, and interneurons including choline acetyltransferase (ChAT)-positive and parvalbumin (PV)-positive cells[40]. All these types of neurons receive

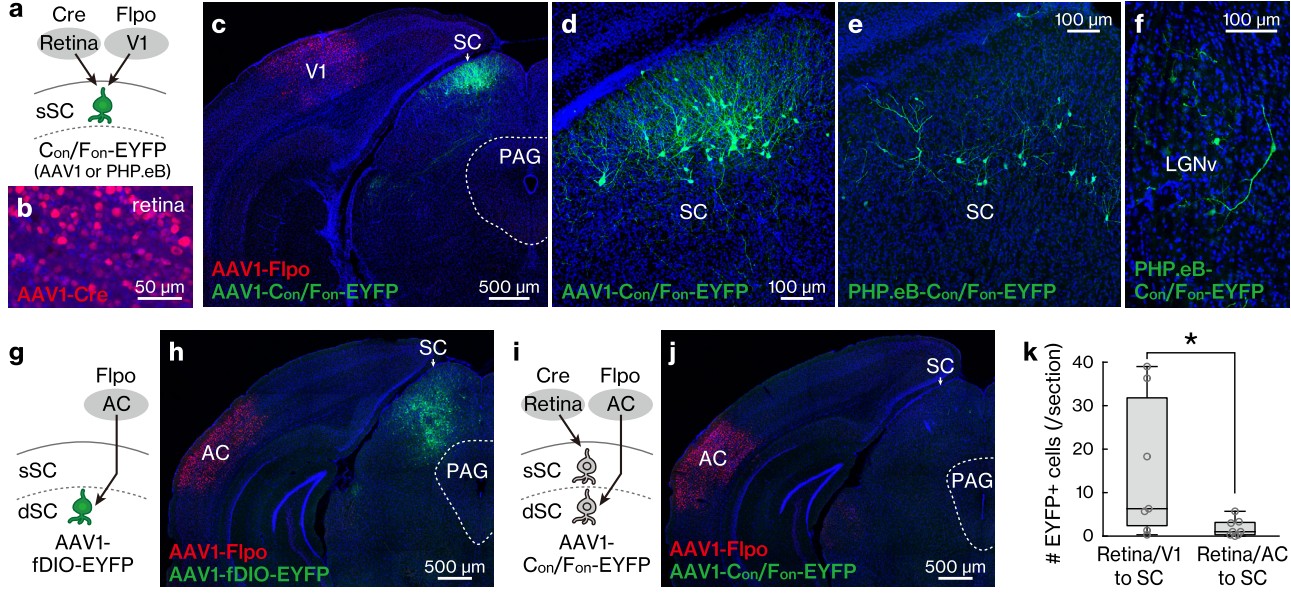

**Fig. 3 Intersectional, anterograde transsynaptic labelling of the retina/V1 to sSC pathway. a–f** Intersectional, anterograde transsynaptic labelling of the retina/V1 to SC pathway. **a** Schematic of the experiment. **b** Representative Cre (red) expression in the retina infected with AAV1-Cre. **b–f**, **h**, **j** Blue, DAPI. **c**, **d** Coronal sections showing the Flpo expression (**c**, red) in the V1 and the EYFP (green)-positive cells in the SC (**c**, **d**). The AAV1 containing $C_{on}/F_{on}$-EYFP expression cassette was injected locally in the SC. $N = 7$ mice. **d** Magnified view of (**c**). **e**, **f** Coronal sections showing the EYFP (green)-positive cells in the left SC (7.0 ± 9.0 cells/section) (**e**) and left LGNv (1.8 ± 1.7 cells/section) (**f**). The $C_{on}/F_{on}$-EYFP expression cassette was delivered to the whole brain by the retroorbital injection of PHP.eB. $N = 4$ mice. **g**, **h** Anterograde transsynaptic labelling of the AC to deep SC (dSC) pathway. **g** Schematic of the experiment. **h** Coronal section showing the EYFP (green) expression in the deep SC following the AAV1-Flpo (red) and AAV1-fDIO-EYFP injections into the AC and SC, respectively. $N = 8$ mice. **i**, **j** Lack of intersectional, anterograde transsynaptic labelling for the retina/AC to SC pathways. **i** Schematic of the experiment. **j** Coronal section showing few EYFP (green)-positive cells in the SC following AAV1-Cre, AAV1-Flpo (red), and AAV1-$C_{on}/F_{on}$-EYFP injections into the retina, AC, and SC, respectively. $N = 8$ mice. **k** Numbers of EYFP-positive cells in the SC for the retina/V1 to SC (**c**, **d**) and retina/AC to SC (**j**) pathway experiments. $N = 7$ (retina/V1 to SC) and 8 (retina/AC to SC) mice. *$P = 0.029$, Mann–Whitney $U$ test. Center line, median; box limits, upper and lower quartiles; whiskers, min and max. sSC superficial superior colliculus, V1 primary visual cortex, LGNv ventral lateral geniculate nucleus, EYFP enhanced yellow fluorescent protein, AC auditory cortex, AAV1 adeno-associated virus serotype 1.

monosynaptic inputs from M2[41–43]. However, it remains unknown whether these individual DS neurons receive bilateral M2 monosynaptic inputs. To test this, we locally injected AAV1-Cre, AAV1-Flpo, and AAV1-hSyn-$C_{on}/F_{on}$-EYFP into the left M2, right M2, and bilateral DS, respectively (Fig. 4a). Cre and Flpo expression in the injected regions was confirmed by immunohistochemistry (Fig. 4b). In the DS, many EYFP-positive cells were observed bilaterally (Fig. 4c–e). We found that the EYFP-labelled DS cells included DARPP-32-positive (mean ± s.d., 49.3 ± 9.6% of the EYFP-positive cells), ChAT-positive (16.1 ± 4.1%), and PV-positive (17.1 ± 2.3%) neurons (Fig. 4f–h). These results suggest that all types of DS neurons examined included cells that received monosynaptic inputs from the bilateral M2.

## Discussion

By combining the anterograde and transneuronal spread of AAV1 with intersectional gene expression, we established a method to genetically label neurons that receive monosynaptic inputs from two upstream regions with a single gene of interest. An anatomical control experiment suggested that this labelling exhibited synaptic specificity. We showed the success of this labelling method in two distinct circuits. We also demonstrated that the systemic delivery of the $C_{on}/F_{on}$ cassette is useful for the unbiased detection of neurons throughout the brain. Recently, the synaptic specificity of AAV1-based transneuronal labelling has been intensively characterised and established using anatomical, electrophysiological, and molecular approaches[14]. In addition, AAV1 shows transsynaptic spread in various pathways, including glutamatergic and GABAergic synapses[14]. Our method provides a

powerful means for determining the locations, numbers, and cell types of neurons receiving monosynaptic inputs from the two defined upstream regions.

This study provides anatomical evidence for the existence of cells receiving interregional, convergent synaptic inputs for both the retina/V1-SC and bilateral M2-DS pathways, indicating that single postsynaptic neurons can integrate information directly received from two distinct upstream regions in these pathways. Regarding the retina/V1-SC pathway, although the retina and V1 are reported to form synapses on SC neurons with similar morphology, it remains unknown whether these neuronal populations are identical[13]. Our results indicate that the retina and V1 share the same or at least overlapping postsynaptic sSC neurons. This anatomical finding further clarifies how information is processed in the SC. SC neurons inherit feature selectivity from the retina[44], while the top-down input from the V1 modulates the gain of visual responses in SC neurons[45]. The circuit mechanism that integrates such retina- and V1-derived information in the SC is unknown. The convergent synaptic inputs on sSC neurons suggest that this information integration takes place in individual SC neurons. Regarding the M2-DS projection, it is involved in coordinated action selection, and its disturbance is associated with uncoordinated involuntary movements in the Huntington's disease model[33,46,47]. Thus, the convergent input from bilateral M2 to single DS neurons may play a role in bilaterally coordinated action selection and movements.

Neurons that receive monosynaptic inputs from two upstream areas have been recently labelled using the combined introduction of AAV1-Cre, AAV1-Flp, Cre-dependent, and Flp-dependent expression cassettes in mice[14,26]. In these studies, the

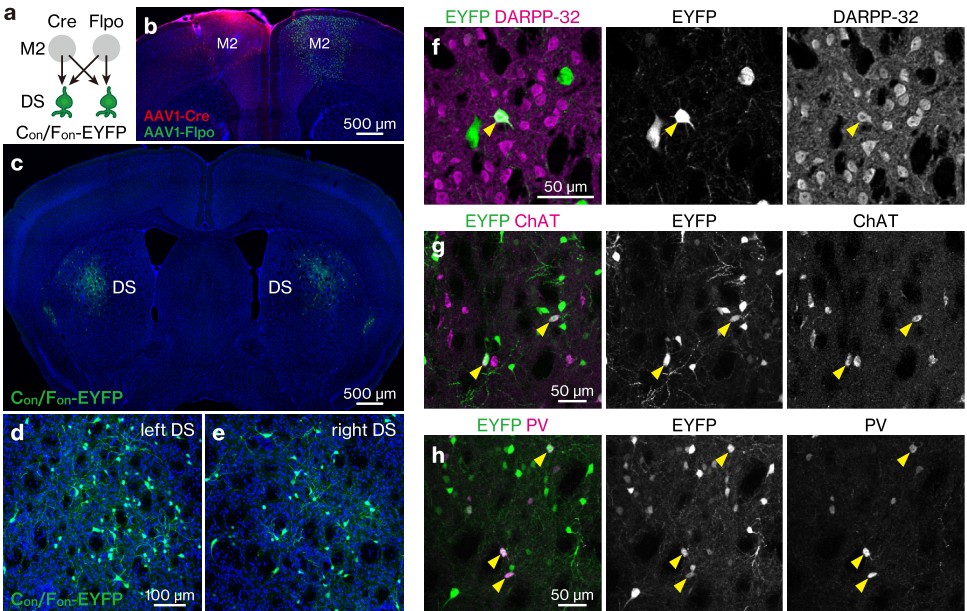

**Fig. 4 Intersectional, anterograde transsynaptic labelling of the bilateral M2 to DS pathway. a** Schematic of the experiment. **b** Cre (red) and Flpo (green) expressions in the left and right M2, respectively. **b–e** All images are coronal sections. Blue, DAPI. $N = 4$ mice. **c–e** EYFP (green) expression in the bilateral DS. **d, e** Magnified views of the left (**d**) and right (**e**) DS. The number of EYFP-positive cells in the DS was $58.6 \pm 36.1$ cells/section/hemisphere. **f–h** Coronal DS sections showing immunohistochemistry for dopamine- and cAMP-Regulated Phosphoprotein 32 kDa (DARPP-32) (**f**), choline acetyltransferase (ChAT) (**g**), or parvalbumin (PV) (**h**). Arrowheads, EYFP (green)-expressing cells double positive for the marker proteins (purple). M2 secondary motor cortex, DS dorsal striatum, AAV1 adeno-associated virus serotype 1, EYFP enhanced yellow fluorescent protein.

postsynaptic neurons of interest were identified as cells that co-express two fluorescent proteins with different colours. Instead, we used the $C_{on}/F_{on}$ intersectional expression cassette, which allowed the specific labelling of the cells with a single gene of interest. Although the use of the $C_{on}/F_{on}$ cassette is straightforward, this approach reduces the number of colour channels needed to specifically identify target neurons, which is advantageous in characterising cell types using multicolour immunohistochemistry and in-situ hybridisation. Moreover, our approach will be particularly useful for selectively manipulating and monitoring target cells with optogenetics, chemogenetics, imaging, and optogenetics-combined extracellular recordings[10]. Thus, our approach expands the application of intersectional anterograde transsynaptic tracing, allowing the investigation of the cell type and function of neurons with convergent synaptic inputs. The intersectional expression system has been progressively expanded in several directions, including the use of mutually exclusive VCre recombinase[16,48], the tetracycline-inducible expression system[49], and split-Cre complementation[50]. Combining these systems with the present method may enable us to label and study neurons that receive monosynaptic inputs from three or more upstream regions. Such intricately integrating 'hub' neurons may play a key role in generating yet to be identified information streams in neural circuits.

A potential drawback of our approach is the necessity for accurate local triple AAV injection. However, the combination with intravenous PHP.eB injection reduced the number of necessary local injections to two, and such double injection is routinely used in many studies. Moreover, several recent studies applied local triple[51–53] or even quadruple[54,55] virus/tracer injections, suggesting that multiple injections are becoming common for the investigation of precise circuit wiring. One of the limitations of AAV1-mediated transsynaptic tagging is that AAV1, similar to many other AAV serotypes, exhibits retrograde transport from axons[13,14,27,28,56,57]. Therefore, the application of this method is restricted to unidirectional pathways with no back projection[13,14]. The mechanisms of directional transport remain unclear. However, the versatility of the AAV1-mediated transsynaptic tagging to dissect specific neurons in reciprocally connected circuits can be enhanced with significant improvements in the ratio of anterograde transsynaptic transport to retrograde transport via capsid engineering[28,38]. Another limitation of this method involves the potential cytotoxicity of Cre. Highly expressed Cre can damage cells by recognising DNA sequences that resemble *loxP* sites[58–60], whereas a relatively high AAV1 titre is necessary for its anterograde transsynaptic spread[13]. This toxicity may be minimised by reducing Cre expression to a moderate level, for instance, by removing the woodchuck hepatitis virus post-transcriptional regulatory element (WPRE) from the AAV transfer plasmid, while maintaining a high AAV1 titre[14]. Although these limitations need to be addressed, our method provides a straightforward approach to genetically dissect neurons that directly receive various types of information from multiple brain regions.

## Methods

**Ethical declarations**. All procedures related to animal care and use were approved by the Institutional Animal Care and Use Committee of Osaka City University (approved protocol #15030) and were performed in accordance with the *Guide for the Care and Use of Laboratory Animals* published by the National Institutes of Health.

**Plasmids**. The following plasmids were obtained from Addgene: pAAV-hSyn-Cre-WPRE (no. 105553), pAAV-EF1α-mCherry-IRES-Flpo (no. 55634), pAAV-EF1α-DIO-EYFP (no. 20296), pAAV-EF1α-fDIO-EYFP (no. 55641), pAAV-hSyn-$C_{on}$/$F_{on}$-EYFP (no. 55650), and pUCmini-iCAP-PHP.eB (no. 103005). hSyn and EF1α denote the human synapsin 1 promoter and elongation factor 1 alpha promoter, respectively. The DIO, fDIO, and $C_{on}$/$F_{on}$ denote Cre-dependent, Flpo-dependent, and Cre/Flpo double-dependent gene expression cassettes, respectively[15,61]. pXR1 and pHelper were obtained from the National Gene Vector Biorepository[62] and the AAV Helper-Free System (no. 240071, Stratagene), respectively. During the AAV1 preparation described below, we were unable to obtain a high titre with pAAV-EF1α-mCherry-IRES-Flpo, which has a long insert (~5.0 kb) exceeding the AAV packaging limit of ~4.4 kb between inverted terminal repeats (ITRs). Therefore, we constructed a shorter plasmid with a ~3.0 kb insert between ITRs, pAAV-hSyn-Flpo-3×FLAG

(no. 173047, Addgene), which has the hSyn promoter followed by Flpo with 3×FLAG C-terminal tag (OGS629, Sigma-Aldrich). This plasmid was verified by sequencing and used throughout the study. Site-specific recombination of the DIO-, fDIO-, and $C_{on}/F_{on}$-EYFP cassettes by Cre and/or Flpo was confirmed by transfecting the corresponding plasmids (0.26 μg/well) into 293T cells (RCB2202, Riken BRC) seeded on 24-well tissue culture plates.

**AAV vector preparation**. AAV vectors were prepared as described previously with minor modifications[25,57,63]. Briefly, AAV1 was produced by co-transfecting three plasmids (pXR1, pHelper, and the pAAV plasmid containing the gene of interest; 20 μg/dish each) using PEI MAX (no. 24765-1, Polysciences) into 293T cells seeded on eight 15-cm tissue culture dishes. For AAV-PHP.eB[38] production, the pXR1 plasmid was switched to the pUCmini-iCAP-PHP.eB plasmid (20 μg/dish). The cells were harvested 72 h after transfection, and AAV was purified using the AAVpro Purification Kit Maxi (All Serotypes) (no. 6666, Takara Bio) according to the manufacturer's instructions. The titre measurement, performed using qPCR (StepOnePlus, Applied Biosystems), included the following: AAV1-Cre $(1.8–2.3 \times 10^{13}$ vg/mL), AAV1-Flpo $(3.9 \times 10^{13}$ vg/mL), AAV1-DIO-EYFP $(1.4 \times 10^{13}$ vg/mL), AAV1-fDIO-EYFP $(1.3 \times 10^{13}$ vg/mL), AAV1-$C_{on}/F_{on}$-EYFP $(1.7 \times 10^{13}$ vg/mL), and PHP.eB-$C_{on}/F_{on}$-EYFP $(1.0 \times 10^{12}$ vg/mL).

**Surgery**. Three types of AAV injections were performed using male C57BL/6 J mice (age on the day of the first surgery, 7.9–13.0 weeks; weight, 19.9–28.6 g; SLC, Japan) under anaesthesia (intraperitoneal injection of a mixture of 0.3, 4, and 5 mg/kg of medetomidine hydrochloride, midazolam, and butorphanol tartrate, respectively)[64]. Intravitreal injection[65] was performed on the right eye to deliver AAV1-Cre into the retina. First, a small incision was made into the sclera using a 29-gauge needle (SS-05M2913, Terumo). Then, a 33-gauge syringe (no. 80008, Hamilton) was inserted through the same incision to inject 1 μL of AAV1-Cre supplemented with 1:10 (v/v) 1% Fast Green FCF (no. 061-00031, Fujifilm Wako Chemicals) at a speed of 3–6 μL/min into the vitreous. The needle was kept inside the vitreous for 1–2 min before removal. Retroorbital injection[66] was performed for the systemic delivery of AAV-PHP.eB. A 29-gauge syringe (SS-05M2913, Terumo) was inserted into the right retroorbital sinus, and 10 μL of PHP.eB-$C_{on}/F_{on}$-EYFP (mixed with 40 μL of 1% Fast Green FCF and 50 μL of phosphate-buffered saline [PBS]) was slowly injected. Stereotaxic injections were performed to deliver AAV to specific brain areas. A craniotomy was performed above the injection site, and AAV was injected (5 μL/h) through a pulled glass pipette. Different AAVs were injected separately on different days with an interval of six or more days in the order of AAV1-Cre, AAV1-Flpo, and one of DIO-, fDIO-, or $C_{on}/F_{on}$-EYFP. This multi-step stereotaxic injection was used to minimise any risk of contamination among the AAVs during surgery[13,25]. The coordinates[67] and volumes are listed as follows: left V1 (anteroposterior from bregma [AP], −3.9 mm; mediolateral from the midline [ML], 2.6 mm; dorsoventral from the cortical surface [DV], 0.6 mm; 0.2–0.4 μL), left AC (AP, −3.1 mm; ML, 4.0 mm; DV, 0.75 mm; 0.2 μL), left SC (AP, −3.9 mm; ML, 0.8 mm; DV, 1.5 mm; 0.4 μL), M2 (AP, 1.5 mm; ML, 0.8 mm; DV, 0.8 mm; 0.2–0.4 μL/region), and DS (AP, 0.5 mm; ML, 2.3 mm; DV, 3.0 mm; 0.4 μL/region).

**Histology**. Mice were transcardially perfused with 0.9% saline and then 4% paraformaldehyde in 0.1 M phosphate buffer under anaesthesia after 18 ± 4 days following the last surgery. The brains and the eyes (for mice with intravitreal injection) were enucleated and stored in the same fixative overnight at 4 °C. The brains were then transferred and kept in 30% sucrose in PBS for more than 48 h at 4 °C and sectioned using a freezing microtome (SM2010R, Leica; EF-13, Nihon Microtome Laboratory) at a thickness of 40 μm parallel to the coronal plane. Whole-mount retinae were prepared by removing the cornea, sclera, and lens from the eyes and making radial incisions into the retina[68]. The sections and retinae were incubated sequentially with 5% bovine serum albumin (BSA)/0.3% Triton X-100 in PBS for either 30 min (for sections) or 60 min (for retinae) at room temperature, primary antibodies in 5% BSA/PBS overnight at 4 °C, and corresponding secondary antibodies with DAPI (0.5 μg/mL, D1306, Thermo Fisher) in 5% BSA/PBS either for 2 h at room temperature or overnight at 4 °C. The samples were washed with PBS between incubations. The following primary antibodies were used: mouse anti-Cre recombinase (1:2,000, MAB3120, Millipore), rabbit anti-FLAG (DYKDDDDK) (1:1,000, RFLG-45A-Z, ICL), chicken anti-GFP (1:2,000, ab13970, Abcam; only for sections with PHP.eB-$C_{on}/F_{on}$-EYFP), rabbit anti-DARPP-32 (1:800, 2306 S, Cell Signaling Technology), goat anti-ChAT (1:100, AB144P, Millipore), and rabbit anti-PV (1:4,000, PV27, Swant). The following secondary antibodies (all from Thermo Fisher) were used at a dilution of 1:800: goat anti-chicken IgY conjugated with Alexa Fluor 488 (A-11039), goat anti-mouse IgG with Alexa Fluor 594 (A-11032), goat anti-rabbit IgG with Alexa Fluor 594 (A-11037), donkey anti-goat IgG with Alexa Fluor 594 (A-11058), and goat anti-rabbit IgG with Alexa Fluor 647 (A-21245). The sections and retinae were mounted on coverslips with an antifade mountant (P36961, Thermo Fisher), and fluorescent images were obtained using confocal microscopes (LSM800 and LSM700, Zeiss) equipped with 10× (numerical aperture = 0.45) and 20× (numerical aperture = 0.8) objectives. Cells positive for EYFP, ChAT, or PV were detected from the fluorescent images using automated software (Image-based Tool for Counting Nuclei, Center for Bio-image Informatics at University of California, Santa Barbara), followed by manual correction of erroneous detections[25]. DARPP-32-positive cells were detected manually using the Fiji software (ver. 2.1.0)[69].

**Statistics and reproducibility**. Statistical analysis was performed using IBM SPSS Statistics (ver. 21, IBM). $P < 0.05$ obtained using the Mann–Whitney $U$ test was considered as a significant difference. All data are reported as the mean ± standard deviation unless specified otherwise.

**Reporting summary**. Further information on research design is available in the Nature Research Reporting Summary linked to this article.

## Data availability
The pAAV-hSyn-Flpo-3×FLAG plasmid is available from Addgene (no. 173047). Data underlying Fig. 3k are presented in Supplementary Data 1. All other data supporting this study are available from the corresponding authors upon reasonable request.

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

## Acknowledgements

We thank the Research Support Platform, Osaka City University Graduate School of Medicine for assistance with confocal microscopy. This work was supported by AMED PRIME (21gm6510003h0001 to T.K.), JST PRESTO (JPMJPR1882 to T.K.), JSPS KAKENHI (20H03356 and 17K19462 to K.M. and 21H05831, 20K06878, and 19H04937 to T.K.), Takeda Science Foundation (to K.M.), The Uehara Memorial Foundation (to K.M. and T.K.), The Naito Foundation (to K.M. and T.K.), Kato Memorial Bioscience Foundation (to T.K.), and Osaka City University Strategic Research Grant for Young Researchers (to T.K.).

## Author contributions

T.K. and K.M. conceived the project. T.K., M.T. and N.K. performed the experiments. T.K. and K.M. wrote the manuscript with input from all the authors.

## Competing interests

The authors declare no competing interests.
