## [Peer Review File · Communications Biology]

Reviewers' comments:

Reviewer #1 (Remarks to the Author):

In this paper Kitanishi et al describe an approach for identification of neurons that receive convergent inputs from two discrete neural pathways. Their method uses an existing dual-recombinase controlled viral expression system, which permits intersectional transgene expression in neurons that contain the recombinases Cre and Flpo (INTRSECT). This cassette expresses a fluorescent reporter only when both Cre and Flpo are expressed. The Authors exploit the trans-synaptic trafficking of viral vectors that express both Cre and Flpo, which has previously been observed when expressed by serotype 1 adeno-associated viral vectors (AAV1). At the heart of their paper is the idea that neurons that receive synaptic input from two pathways can be identified by expressing the INTRSECT cassette in those neurons and by separately delivering Cre and Flpo by AAV1 vectors delivered to upstream neurons.

The authors validate the dual recombinase dependence of the reporter cassette in vitro and show that, individually, AAV1-Cre and AAV1-Flpo vectors can be used to control Cre- and Flpo-dependent recombination in the tectal visual pathway, a system that is convenient due to the unidirectional nature of the retina-colliculus pathway. They then show that the INTRSECT vector can be used to label neurons in the superior colliculus that are putative recipients of input from both retinal and primary visual cortex (V1) neurons. Interestingly, they also show that their approach can be used to identify global sites of convergence when the INTRSECT vector is delivered systemically.

The Authors also show the applicability of the approach to a number of other pathways: they show that few SC neurons receive convergent input from both retina and auditory cortex (as expected: most retinal inputs terminate in the superficial SC whereas most auditory cortex inputs terminate in the deep SC – this is interpreted as anatomical validation of the monosynaptic dependence of the system), and that all major subgroups of striatal neurons receive bilateral input from the secondary motor cortex.

The Authors apply a number of previously developed tools in a new way: dual-recombinase and even triple-recombinase vectors and transgenic animals are widely available, permitting ever more precise targeting of neuronal subtypes based on genetic or morphological criteria (Madisen et al. (2015) doi:10.1016/j.neuron.2015.02.022, Fenno et al. (2014) doi:10.1038/nmeth.2996). Similarly, the trans-synaptic trafficking of AAV1 mediated recombinases is now a commonly used approach for gaining genetic access to post-synaptic network constituents: this approach, sometimes called 'Cre-tagging', came to prominence as a result of the work by Zingg et al (2017, doi:10.1016/j.neuron.2016.11.045), which described post-synaptic Cre-dependent transgene expression in the mouse retina-colliculus pathway (inter alia) when Cre is delivered via AAV vectors (especially AAV1). That approach was recently extended by the same group, showing similar trans-synaptic delivery of AAV1-Flpo (Zingg et al. (2020). doi:10.1523/JNEUROSCI.2158-19.2020). The idea to use two AAV1 vectors, driving Cre and Flpo, to identify sites of convergence, revealed by conditional reporter expression in a third vector, has also been demonstrated previously (Oh et al. (2020) doi:10.5607/en20006), although they used a single vector in which different coloured reporters were individually controlled by Cre/Flpo.

As such, although the work described here is of high quality, the data convincing, and the figures beautiful, the novelty of the concept is limited to the decision to use a vector that drives reporter expression conditional on the presence of Cre AND Flpo. As the work has been framed around the technical approach used, rather than the biological insights gained from it, to my mind that limits to the novelty and likely impact of the paper and the extent to which it can be improved by revision. I feel it probably does not meet the Journal's stated aim of publishing "significant advances bringing new biological insight to a specialized area of research".

The approach described is likely to be relevant to researchers who wish to identify sites of convergence of pathways of interest, although the necessity for conducting multiple accurate viral microinjections is a drawback. The paper is well written and appropriately covers the key literature in this field.

Reviewer #2 (Remarks to the Author):

The study by Kitanishi et al. tested the intersectional application of transsynaptic AAV1 methods. My major concern with the work is that the proposed method has been already well demonstrated in the literature, and that there is barely any new information generated from the study.

Fig.1 This demonstration is completely unnecessary. Similar application of Con/Fon has been widely used in the field.

Fig.2. AAV1-Cre and AAV1-Flpo in anterograde transneuronal spread have been already demonstrated in the literature. This figure does not generate any new information.

Fig.3. Similar intersectional application of AAV1-cre and AAV1-flp is used in Zingg etc. 2020, although for different purpose. Results on V1-retina or AC-retina are at most confirmatory.

Fig.4. Same SC neurons receive both ipsi and contralateral inputs from M2 are potentially interesting, but to be a major claim, it's very preliminary.

Statistics and quantification for all the experiments are very limited.

Reviewer #3 (Remarks to the Author):

Kitanishi et al. describe an intersectional approach for accessing neurons that are defined by their convergent input from two upstream brain regions. The authors take advantage of the recently characterized anterograde transsynaptic spread of AAV1 and use injections of AAV1-Cre and AAV1-Flpo to transduce neurons downstream of each injection site. Cells that co-express both Cre and Flpo, and thus likely receive convergent synaptic input from both regions, are then identified via fluorescent expression of a Cre & Flpo-dependent AAV. This is either locally injected into a target region, or systemically delivered for brain-wide detection of Cre+/Flpo+ cells. The authors demonstrate this method in two circuits: V1/Retina -> SC and M1-contra/ipsi -> Str. Both pathways meet the important requirement of being unidirectional, as potential retrograde spread of AAV1 confounds the interpretation of cell labeling in reciprocally connected regions (noted in their discussion). In addition, the choice of using Cre and Flpo for conditional expression helps rule out any potential concerns related to cross-reactivity between each recombinase, as these have been widely reported to exhibit high specificity for LoxP and FRT sites, respectively (e.g. Madisen L et al., Neuron, 2015). The authors provide a nice demonstration of this selectivity in Fig.1B,C and directly confirm the lack of cross-reactivity.

Overall, the experiments are carefully designed and well executed for each of the demonstrated pathways. The resulting insight gained from the outcome of each of the experiments is somewhat limited, though. In particular, when most of the cell-types within a target region appear to receive input from both upstream sources (e.g. Fig.4), it remains unclear as to whether this reflects the true pattern of synaptic connectivity or whether some amount of extrasynaptic viral spread may account for this labeling result. Ideally, there would be some circuit that could be tested using this approach where a specific subset of cells are expected to be co-labeled and other intermingled cell-types are excluded. However, aside from previous demonstrations in the cerebellum (Fig.1 of Zingg et al., 2020), I am unaware of any other unidirectional pathways that meet this criteria. That being said, the technique described here provides a valuable means to continue exploring these questions in many different circuits. The authors clearly demonstrate the utility and feasibility of their new method, and given the increasing use of AAV1 for anterograde transsynaptic circuit studies, I believe the approach outlined here will be of broad interest and applicability to the neuroscience field.

RESPONSE TO REVIEWERS

We thank the reviewers for their valuable comments. We have revised our manuscript thoroughly in response to their suggestions. In response to the comments from Reviewer 1, we specified the novelty of the present study, emphasized the biological significance gained from our results, and discussed the necessity for multiple viral microinjections in the Discussion. In response to the comments from Reviewer 2, we described the relevance of Figures 1 and 2, specified the novelty and the biological significance of the present study in the Discussion, and added quantitative data for EYFP-positive cells to the figure legends. In response to the comments from Reviewer 3, we discussed the technical challenges of the rigorous measurement of extrasynaptic viral spread.

Below, the reviewers' comments are shown in bold Arial font and our responses in regular Times New Roman font. Text added to or changed in the revised manuscript is in blue font, both here and in the manuscript.

Reviewer #1 (Remarks to the Author):

Comment: In this paper Kitanishi et al describe an approach for identification of neurons that receive convergent inputs from two discrete neural pathways. Their method uses an existing dual-recombinase controlled viral expression system, which permits intersectional transgene expression in neurons that contain the recombinases Cre and Flpo (INTRSECT). This cassette expresses a fluorescent reporter only when both Cre and Flpo are expressed. The Authors exploit the trans-synaptic trafficking of viral vectors that express both Cre and Flpo, which has previously been observed when expressed by serotype 1 adeno-associated viral vectors (AAV1). At the heart of their paper is the idea that neurons that receive synaptic input from two pathways can be identified by expressing the INTRSECT cassette in those neurons and by separately delivering Cre and Flpo by AAV1 vectors delivered to upstream neurons.

The authors validate the dual recombinase dependence of the reporter cassette in vitro and show that, individually, AAV1-Cre and AAV1-Flpo vectors can be used to control Cre- and Flpo-dependent recombination in the tectal visual pathway, a system that is convenient due to the unidirectional nature of the retina-colliculus pathway. They then show that the INTRSECT vector can be used to label neurons in the superior colliculus that are putative recipients of input from both retinal and primary visual cortex (V1) neurons. Interestingly, they also show that their approach can be used to identify global sites of convergence when the INTRSECT vector is delivered systemically.

The Authors also show the applicability of the approach to a number of other pathways: they show that few SC neurons receive convergent input from both retina and auditory cortex (as expected: most retinal inputs terminate in the superficial SC whereas most auditory cortex inputs terminate in the deep SC – this is interpreted as anatomical validation of the monosynaptic dependence of the system), and that all major subgroups of striatal neurons receive bilateral input from the secondary motor cortex.

The Authors apply a number of previously developed tools in a new way: dual-recombinase and even triple-recombinase vectors and transgenic animals are widely available, permitting ever more precise targeting of neuronal subtypes based on genetic or morphological criteria (Madisen et al. (2015) doi:10.1016/j.neuron.2015.02.022, Fenno et al. (2014) doi:10.1038/nmeth.2996). Similarly, the trans-synaptic trafficking of AAV1 mediated recombinases is now a commonly used approach for gaining genetic access to post-synaptic network constituents: this approach, sometimes called 'Cre-tagging', came to prominence as a result of the work by Zingg et al (2017, doi:10.1016/j.neuron.2016.11.045), which described post-synaptic Cre-dependent transgene expression in the

mouse retina-colliculus pathway (inter alia) when Cre is delivered via AAV vectors (especially AAV1). That approach was recently extended by the same group, showing similar trans-synaptic delivery of AAV1-Flpo (Zingg et al. (2020). doi:10.1523/JNEUROSCI.2158-19.2020). The idea to use two AAV1 vectors, driving Cre and Flpo, to identify sites of convergence, revealed by conditional reporter expression in a third vector, has also been demonstrated previously (Oh et al. (2020) doi:10.5607/en20006), although they used a single vector in which different coloured reporters were individually controlled by Cre/Flpo.

As such, although the work described here is of high quality, the data convincing, and the figures beautiful, the novelty of the concept is limited to the decision to use a vector that drives reporter expression conditional on the presence of Cre AND Flpo. As the work has been framed around the technical approach used, rather than the biological insights gained from it, to my mind that limits to the novelty and likely impact of the paper and the extent to which it can be improved by revision. I feel it probably does not meet the Journal's stated aim of publishing "significant advances bringing new biological insight to a specialized area of research".

Reply: We thank the reviewer for the valuable comments. As pointed out, the conceptual novelty of the present study is the use of the C_{on}/F_{on} cassette together with AAV1-Cre and AAV1-Flpo. Although this is a simple concept, it allows to genetically target neurons with a single gene of interest, but not with two fluorescent proteins, as reported in previous studies (Ref #14; Zing et al., J Neurosci, 2020; ref #26; Oh et al., Exp Neurol, 2020). Thus, our approach reduces the number of color channels necessary for identifying target neurons and is, thus, advantageous for characterizing cell types using multicolor immunohistochemistry and *in-situ* hybridization. Moreover, our approach can be readily applied to the selective manipulation and monitoring of target cells with various techniques (e.g., optogenetics, chemogenetics, imaging, and

optogenetics-combined extracellular recordings), which is otherwise not possible. We believe that this is a key conceptual advancement that enables others to investigate the cell type and function of neurons with convergent synaptic inputs. We specified this point in the corresponding Discussion section.

Regarding the biological insights, we have substantially revised the Discussion section to emphasize the biological significance of the present study for both retina/V1-SC and M2-DS pathways. As for the retina/V1-SC pathway, although the retina and V1 are known to form synapses on SC neurons with similar morphology, it remains unclear whether these neuronal populations are identical (ref #14; Zingg et al., *Neuron*, 2017). Our results in Figure 3 indicate that the retina and V1 share the same or at least overlapping postsynaptic SC neurons. This anatomical finding further implies how information is processed in the SC. SC neurons inherit feature selectivity from the retina (ref #44; Shi et al., *Nat Neurosci*, 2017), whereas the top-down input from the V1 modulates the gain of visual responses in SC neurons (ref #45; Zhao et al., *Neuron*, 2014). The circuit mechanism to integrate such retinal and V1 information in the SC is unknown. The convergent synaptic inputs on SC neurons described in the present study suggest that this integration can take place in single SC neurons.

Similarly, the convergent synaptic inputs in the bilateral M2 to DS pathway might be essential for information integration in single DS neurons. The M2-DS pathway is involved in coordinated action selection, and the disturbance of this pathway is associated with uncoordinated involuntary movements in the Huntington's disease model (ref #46; Sul et al., *Nat Neurosci*, 2011; ref #33; Hintiryan et al., *Nat Neurosci*, 2016; ref #47; Fernández-García et al., *eLife*, 2020). Thus, the convergent input from bilateral M2 to single DS neurons (Figure 4) may play a key role in bilaterally coordinated action selection and movements. In summary, the present study provides anatomical evidence for convergent synaptic inputs to individual postsynaptic neurons in two pathways, which further implicates how information is processed in the local circuit of postsynaptic areas.

Comment: The approach described is likely to be relevant to researchers who wish to identify sites of convergence of pathways of interest,

although the necessity for conducting multiple accurate viral microinjections is a drawback. The paper is well written and appropriately covers the key literature in this field.

Reply: We thank the reviewer for raising this point. As pointed out, the present approach generally requires triple AAV microinjection. However, the combination of intravenous PHP.eB injection reduces the number of necessary microinjections to two (Figures 4e-f), and such double injection is commonly used in many studies. Moreover, several recent studies applied triple (ref #51; Schwarz et al., Nature, 2015; ref #52; Zingg et al., J Comp Neurol, 2018; ref #53; Foster et al., Nature, 2021) or even quadruple (ref #54; Bienkowski et al., Nat Neurosci, 2018; ref #55; Marriott et al., J Comp Neurol, 2021) virus/tracer microinjections, suggesting that multiple injections are becoming more common to precisely investigate circuit wiring. Thus, we believe that double/triple microinjection is not a critical drawback that hinders the utility of the present approach. We added the above discussion to the corresponding Discussion section.

Reviewer #2 (Remarks to the Author):

Comment: The study by Kitanishi et al. tested the intersectional application of transsynaptic AAV1 methods. My major concern with the work is that the proposed method has been already well demonstrated in the literature, and that there is barely any new information generated from the study.

Fig.1 This demonstration is completely unnecessary. Similar application of Con/Fon has been widely used in the field.

Reply: We agree with the reviewer that C_{on}/F_{on} has been widely used. However, the primary reason for this demonstration was to show that the newly constructed Flpo-3×FLAG shows the expected recombination. A previously published Flpo plasmid,

pAAV-EF1 α -mCherry-IRES-Flpo (no. 55634, Addgene), has a long insert that exceeds the AAV packaging limit, and with this plasmid, we were unable to obtain a high titer AAV1, which is necessary for transsynaptic spread. Thus, we newly constructed a shorter Flpo plasmid with a fused 3 \times FLAG tag, pAAV-hSyn-Flpo-3 \times FLAG, as described in the Methods. Since adding a tag occasionally disturbs molecular functions, we tested and showed in Figure 1b-1c that the Flpo-3 \times FLAG functions as expected (i.e., site-specific recombination and no crosstalk with Cre). We admit that our original manuscript lacked a clear description of why we included Figure 1b-c. We added the following description to the corresponding passage in the Results: “As the Cre- and Flpo-mediated recombination of the C_{on}/F_{on} cassette has been well characterized^{15,16}, this test primarily aimed at verifying the selective recombination with the newly constructed pAAV-hSyn-Flpo-3 \times FLAG plasmid, which expresses 3 \times FLAG-tagged Flpo (see Methods).”

Comment: Fig.2. AAV1-Cre and AAV1-Flpo in anterograde transneuronal spread have been already demonstrated in the literature. This figure does not generate any new information.

Reply: We agree with the reviewer that the anterograde transneuronal spread of AAV1-Cre and AAV1-Flpo has been demonstrated in several pathways. However, whether all individual pathways allow the transneuronal spread of AAV1 remains unclear. Indeed, Zingg et al. 2020 (ref #14) showed large across-pathway differences in the efficiency of transneuronal spread. Thus, in the absence of published evidence, verifying transneuronal spread in the pathway of interest would be crucial. This is particularly important in the present study because our intersectional, dual pathway approach is based on efficient transneuronal spread in single pathways. To our knowledge, Figure 2 is the first demonstration of the transneuronal spread of AAV1-Cre in the M2-DS pathway, and AAV1-Flpo in the retina-SC and M2-DS pathways. Therefore, Figure 2 provides new information and, we believe, is an essential part of the present study. We added the above explanation to the corresponding Results section.

Comment: Fig.3. Similar intersectional application of AAV1-cre and AAV1-flp is used in Zingg etc. 2020, although for different purpose. Results on V1-retina or AC-retina are at most confirmatory.

Reply: Thank you for raising this point. As described in the Discussion, Zingg et al. 2020 (ref #14) and Oh et al. 2020 (ref #26) used AAV1-Cre, AAV1-Flp, Cre-dependent, and Flp-dependent cassettes in single mice. These studies identified neurons receiving convergent inputs as the cells that co-express two different fluorescent proteins. While this approach successfully identifies the cells of interest, it does not readily allow us to manipulate or monitor cellular activity selectively. In contrast, using C_{on}/F_{on} , we targeted neurons with a single gene of interest, which would not only allow for the identification of cells but also the manipulation and measurement of the cellular activity. In addition, the use of C_{on}/F_{on} reduces the number of colour channels necessary for identifying target neurons, which is advantageous for characterizing cell types using multicolour immunohistochemistry and *in-situ* hybridization. Thus, although the use of C_{on}/F_{on} is a simple idea, the present approach expands the application of intersectional anterograde transsynaptic targeting, allowing us to investigate the cell type and function of neurons receiving convergent synaptic inputs. We emphasized this point in the corresponding Abstract, Introduction, and Discussion sections.

Comment: Fig.4. Same SC neurons receive both ipsi and contralateral inputs from M2 are potentially interesting, but to be a major claim, it's very preliminary.

Reply: Figure 4 demonstrates that the present approach is versatile and can be applied to other circuits than that tested in Figure 3. This is why we did not investigate the detailed biological significance of the M2-DS pathway. To specify potential biological insights gained from this experiment, we revised the Discussion section as follows: “Regarding the M2-DS projection, it is involved in coordinated action selection, and its disturbance is associated with uncoordinated involuntary movements in the Huntington’s disease model^{33,46,47}. Thus, the convergent input from bilateral M2 to single DS neurons may

play a role in bilaterally coordinated action selection and movements.”

Comment: Statistics and quantification for all the experiments are very limited.

Reply: We thank the reviewer for raising this point. We quantified the numbers of EYFP-positive cells in postsynaptic areas (i.e., SC or DS) and described them in the legends of Figures 2–4 in the revised manuscript. In Figure 2, we did not perform a statistical test, as the numbers of animals were limited to two or three. In Figures 3 and 4, the added EYFP-positive cell numbers simply state the existence of these cells and are not subject to statistical tests.

Reviewer #3 (Remarks to the Author):

Comment: Kitanishi et al. describe an intersectional approach for accessing neurons that are defined by their convergent input from two upstream brain regions. The authors take advantage of the recently characterized anterograde transsynaptic spread of AAV1 and use injections of AAV1-Cre and AAV1-Flpo to transduce neurons downstream of each injection site. Cells that co-express both Cre and Flpo, and thus likely receive convergent synaptic input from both regions, are then identified via fluorescent expression of a Cre & Flpo-dependent AAV. This is either locally injected into a target region, or systemically delivered for brain-wide detection of Cre+/Flpo+ cells. The authors demonstrate this method in two circuits: V1/Retina -> SC and M1-contralateral/ipsilateral -> Str. Both pathways meet the important requirement of being unidirectional, as potential retrograde spread of AAV1 confounds interpretation of cell labeling in reciprocally connected regions (noted in their discussion). In addition, the choice of using Cre and Flpo for conditional expression helps rule out any potential concerns related to cross-reactivity between each recombinase, as these have been widely reported to exhibit high

specificity for LoxP and FRT sites, respectively (e.g. Madisen L et al., Neuron, 2015). The authors provide a nice demonstration of this selectivity in Fig.1B,C and directly confirm the lack of cross-reactivity.

Overall, the experiments are carefully designed and well executed for each of the demonstrated pathways. The resulting insight gained from the outcome of each of the experiments is somewhat limited, though. In particular, when most of the cell-types within a target region appear to receive input from both upstream sources (e.g. Fig.4), it remains unclear as to whether this reflects the true pattern of synaptic connectivity or whether some amount of extrasynaptic viral spread may account for this labeling result. Ideally, there would be some circuit that could be tested using this approach where a specific subset of cells are expected to be co-labeled and other intermingled cell-types are excluded. However, aside from previous demonstrations in the cerebellum (Fig.1 of Zingg et al., 2020), I am unaware of any other unidirectional pathways that meet this criteria. That being said, the technique described here provides a valuable means to continue exploring these questions in many different circuits. The authors clearly demonstrate the utility and feasibility of their new method, and given the increasing use of AAV1 for anterograde transsynaptic circuit studies, I believe the approach outlined here will be of broad interest and applicability to the neuroscience field.

Reply: We thank the reviewer for the valuable comments. Regarding Figure 4, all cell types examined in the present study (i.e., DARPP-32-, ChAT-, and PV-positive cells) are reported to receive monosynaptic input from M2, as described in the manuscript (ref #41-43). Thus, the labelling of all cell types does not necessarily imply the presence of extrasynaptic AAV1 spread. As pointed out by the reviewer, rigorously measuring the amount of extrasynaptic AAV1 spread is technically challenging. To find convergent pathways suited for probing the synaptic specificity of the present approach, it would be necessary to comprehensively map synaptic connections across intermingled cell types

using other methods than AAV1 (e.g., electron microscopy or patch-clamp recording). We believe that this is beyond the scope of the present study. Although cell populations are not intermingled, the lack of labelling in the retina/AC → SC pathway (Figure 3) suggests that AAV1 does not spread from the passing AC-to-dSC axons to the nearby sSC cells receiving retinal inputs. While the synaptic specificity needs to be rigorously confirmed using other methods in the future, we believe that the present approach provides promising means to analyse the interregional integration of information.

REVIEWERS' COMMENTS:

Reviewer #2 (Remarks to the Author):

The revision and responses are very helpful to clarify many issues and comments.

Reviewer #3 (Remarks to the Author):

The authors have done a good job addressing the reviewer concerns, and I support publication of the revised manuscript.